# Epidemiological characteristics of ischemic heart disease: A comparative study between China and the world from 1990 to 2021 and prediction to 2036

Jiale Wu[1☯], Jinchan Zhai[2☯], Xingyun Zhu[2☯], Quan Hao[2], Jiaqi Yang[2], Xiaoju Li[2,3,4,5], Yunhua Hu[2,3,4,5]*, Qiang Niu[2,3,4,5]*, Yizhong Yan[2,3,4,5]*

**1** Department of Clinical Medicine, School of Medicine, Shihezi University, Shihezi, Xinjiang, China, **2** Department of Preventive Medicine, School of Medicine, Shihezi University, Shihezi, Xinjiang, China, **3** Key Laboratory for Prevention and Control of Emerging Infectious Diseases and Public Health Security, the Xinjiang Production and Construction Corps, Shihezi, Xinjiang, China, **4** Key Laboratory of Preventive Medicine, Shihezi University, Shihezi, Xinjiang, China, **5** Key Laboratory of Xinjiang Endemic and Ethnic Diseases (Ministry of Education), School of Medicine, Shihezi University, Shihezi, Xinjiang, China

☯ These authors contributed equally to this work.
* huyunhua1019@sina.com (YH); niuqiang1022@163.com (QN); erniu19880215@sina.com (YY)

## Abstract

### Background

Ischemic heart disease (IHD) is a major disease that endangers human health. There are relatively few comparative studies on the epidemiological characteristics of IHD.

### Aims

This study aimed to compare and analyze the incidence, prevalence and mortality of IHD in China and the world from 1990 to 2021, predict its change trend from 2022 to 2036, and provide a basis for effective prevention and control of IHD.

### Methods

Data on IHD in China and the world, encompassing age-standardized incidence rate (ASIR), age-standardized prevalence rate (ASPR), and age-standardized mortality rate (ASMR), was extracted from the Global Burden of Disease Study 2021. The Joinpoint regression model was employed to calculate the average annual percentage change (AAPC). The projected data for 2022–2036 were derived through calculations using the Bayesian age-period-cohort model.

### Results

From 1990 to 2021, the growth rates of China's ASIR, ASPR, and ASMR of IHD were 15.97%, 20.42%, and 17.81% respectively, with annual percentage changes of 0.49%, 0.60%, and 0.49%, all higher than global levels; the annual change rates

**Data availability statement:** The datasets for this article are available from the Global Health Data Exchange query tool (http://ghdx.healthda-ta.org/gbd-results-tool).

**Funding:** The Shihezi University High-level Talents Program (RCZK2021B28). The Science and Technology Planning Project of Xinjiang Production and Construction Corps (2023AB049). The Shihezi University self-funded project (ZZZC202125). Tianshan Young Talent Scientific and Technological Innovation Team: Innovative Team for Research on Prevention and Treatment of High-incidence Diseases in Central Asia (2023TSYCTD0020).

**Competing interests:** The authors have declared that no competing interests exist.

of incidence, prevalence, and mortality were 3.17%, 3.23%, and 3.54% respectively, also exceeding the world 1.00%, 1.38%, and 0.43%. In 1990, globally and in China, male incidence, prevalence, and mortality of IHD were generally higher than those of females, except that Chinese female mortality was lower; males had higher mortality but lower prevalence and related annual percentage changes, with the disease burden peaking in people aged 80 years and above. Over the next 15 years, the world and Chinese ASMR of IHD is projected to decline, while the male ASIR and ASPR will decrease, and those of females will increase.

## Conclusions

The burden of IHD and the increase rate in China was higher than global figures. Although ASMR for IHD in China and the world will decline over the next 15 years, IHD remains a public health issue that requires ongoing attention.To reduce the burden of IHD, targeted preventive measures and relevant knowledge popularization should be adopted for populations of different genders and ages.

## 1. Introduction

Ischemic heart disease (IHD) is a kind of atherosclerotic cardiovascular disease (ASCVD). It refers to the heart disease caused by coronary atherosclerosis that narrows or blocks the vascular lumen, or myocardial ischemia, hypoxia or necrosis caused by coronary artery spasm, which poses a serious threat to human life and health [1,2]. In recent years, the burden of IHD in China was on the rise as a whole, not only among people over 70 years old, but also gradually toward younger people. Atherosclerosis can start early in life, and the risk of all-cause death from sub-clinical atherosclerosis increases significantly in young people. About one in seven cardiovascular events in women and one in four in men occur before the age of 55 [3]. It was estimated that the prevalence of ASCVD among young people ranged from 7% to 30% [4,5], a phenomenon that makes the prevention and treatment of IHD even more complex and urgent.

Epidemiological study on the current situation and change trend of IHD will benefit the scientific prevention and control of IHD. Based on the Global Burden of Disease Study 2021 (GBD 2021) database, this study analyzed and compared the characteristics of the incidence, prevalence and mortality of IHD in China and the world from 1990 to 2021, predicted the future trend by gender from 2022 to 2036, and provided epidemiological evidence for the formulation of prevention and control policies for IHD.

## 2. Methods

### 2.1. Data sources

The data related to the burden of IHD were obtained from the GBD 2021 database, which was published by the Center for Health Measurement and Evaluation of the

University of Washington, and provided epidemiological data for 204 countries or regions, 371 diseases, and 88 risk factors [6]. According to the coding principles of the International Classification of Diseases (11th Edition), with data on the incidence, prevalence, and mortality of IHD in China and globally retrieved from GBD 2021, a systematic comparative study was conducted.The data for this study were sourced from a publicly available database, requiring no ethical approval or informed consent. No ethical approval and informed consent were required because of the public availability of GBD and no identifiable information was included in the analyses.

## 2.2. Data analysis

In order to reduce the impact of age structure differences, based on the GBD world standard population [7], standardization was applied to the incidence, prevalence, and mortality of IHD across China and the globe, with this process yielding the age-standardized incidence rate (ASIR), age-standardized prevalence rate (ASPR), and age-standardized mortality rate (ASMR).

The Joinpoint model was used for regression analysis of disease burden indicators, and the average annual percentage of change (AAPC) and its 95% confidence interval (CI) were calculated. For the Joinpoint model implementation, we set the initial number of joinpoints to 0 and defined the range for the number of joinpoints as 0–5. This range is in line with common practices to balance model fit and avoid overfitting, especially considering the temporal scope of our data from 1990 to 2021. The model establishes a segmented regression based on the temporal characteristics of the disease distribution, splits the study time into different intervals through a number of connecting points, and optimizes the trend fitting for each interval. The regression coefficients for each interval were weighted to obtain the AAPC and its 95% CI to estimate the overall change over the study period [7,8], as follows:

$$\ln(\gamma) = \alpha + \beta_i x + \varepsilon$$

$$APCs = 100 \times (\exp(\beta_i) - 1)$$

$$AAPCs = \left\{ exp\left(\sum \omega_i \beta_i / \sum \omega_i\right) \right\} \times 100$$

Where $\gamma$ is ASIR, ASPR or ASMR, $x$ is the year, $\alpha$ is the intercept, $\beta_i$ is the slope coefficient of each time period, $\omega_i$ is the number of years in each time period [9]. When AAPC and its 95% CI are both $>0, <0$, and 0, the corresponding indicators show an upward trend, a downward trend, and remain stable, and a two-sided p-value less than 0.05 was regarded as significant [10].

The Bayesian age-period-cohort (BAPC) model was used to predict the ASIR, ASPR and ASMR of IHD by sex from 2022 to 2036. The future standardized population was predicted by 2021 GBD [11]. It can combine the sample information with the prior information of unknown parameters to obtain the posterior information, and then infer the unknown parameters according to the posterior information [8]. The integration of nested Laplacian approximate BAPC models using BAPC and Integrated Nested Laplace Approximation (INLA) packages in R 4.3.2, which can help policymakers and researchers understand the expected trends of specific diseases in the future, thus providing a basis for the allocation of health resources and the formulation of disease prevention strategies [12].

## 3. Results

### 3.1. The burden of IHD in the world in 1990–2021 and prediction in 2022–2036

**3.1.1. Incidence of IHD in the world.** From 1990 to 2021, the number of new IHD cases worldwide rose from 15,813,600–31,872,800. The growth rate was 101.55%, with an upward trend (AAPC = 2.29%, 95% CI: 2.27%~2.31%,

P<0.001). The incidence rate rose from 296.49 per 100,000 to 403.89 per 100,000, a 36.22% increase, and showed an upward trend (AAPC=1.00%, 95% CI: 0.99%–1.02%, P<0.001). The ASIR decreased from 419.54 per 100,000 to 372.90 per 100,000, a 11.12% decrease, with a downward trend (AAPC=−0.38%, 95% CI: −0.40%~−0.36%, P<0.001). (Table 1)

For 2022–2036 in the world, the ASIR in men is predicted to decrease from 202.60 per 100,000 in 2022 to 178.61 per 100,000 in 2036, and the ASIR in women will rise from 153.89 per 100,000 in 2022 to 165.32 per 100,000 in 2036. (Fig 1)

**3.1.2. Prevalence of IHD in the world.** From 1990 to 2021, the number of global IHD cases rose from 112,169,500–254,276,300. The growth rate was 126.69%, with an upward trend (AAPC=2.67%, 95% CI: 2.65%~2.69%, P<0.001). The prevalence rate rose from 2103.06 per 100,000 to 3222.21 per 100,000, a 36.22% increase, and showed an upward trend (AAPC=1.00%, 95% CI: 1.36%~1.40%, P<0.001). The ASPR rose from 2904.72 per 100,000 to 2946.38 per 100,000, a 1.43% increase, with a downward trend (AAPC=−0.38%, 95% CI: 0.00%~0.06%, P<0.001). (Table 2)

For 2022–2036 in the world, the ASPR in men is predicted to decrease from 1633.03 per 100,000 in 2022 to 1605.57 per 100,000 in 2036, and the ASPR in women will rise from 1222.61 in 2022 per 100,000 to 1451.32 per 100,000 in 2036. (Fig 2)

**3.1.3. Mortality of IHD in the world.** From 1990 to 2021, the number of global deaths rose from 5,367,100–8,991,600. The growth rate was 67.53%, with an upward trend (AAPC=1.72%, 95% CI: 1.54%~1.90%, P<0.001). The mortality rate rose from 100.63 per 100,000 to 113.94 per 100,000, a 13.22% increase, and showed an upward trend (AAPC=0.43%, 95% CI: 0.28%~0.58%, P<0.001). The ASMR decreased from 158.90 per 100,000 to 108.73 per 100,000, a 31.57% decrease, with a downward trend (AAPC=−1.20%, 95% CI: −1.37%~−1.03%, P<0.001). (Table 3)

For 2022–2036, the ASMR of the world in both men and women are predicted to decrease from 119.33 per 100,000 and 72.85 per 100,000 in 2022 to 108.17 per 100,000 and 65.01 per 100,000 in 2036, respectively (Fig 3).

**Table 1. Incidence of IHD in China and the world, 1990-2021.**

| Gender | Number of new cases (10,000) | | Incidence rate (per 100,000) | | ASIR (per 100,000) | |
|---|---|---|---|---|---|---|
| | China | Global | China | Global | China | Global |
| Both | | | | | | |
| 1990 | 230.16 | 1581.36 | 195.64 | 296.49 | 315.31 | 419.54 |
| 2021 | 730.46 | 3187.28 | 513.41 | 403.89 | 365.67 | 372.90 |
| Change (%) | 217.37 | 101.55 | 162.43 | 36.22 | 15.97 | −11.12 |
| AAPC(%) | 3.80* | 2.29* | 3.17* | 1.00* | 0.49* | −0.38* |
| 95%CI | 3.67, 3.92 | 2.27, 2.31 | 3.10, 3.23 | 0.99, 1.02 | 0.43, 0.55 | −0.40, −0.36 |
| Male | | | | | | |
| 1990 | 124.44 | 907.06 | 205.06 | 337.73 | 348.59 | 522.16 |
| 2021 | 381.86 | 1796.16 | 524.46 | 453.65 | 401.19 | 450.39 |
| Change (%) | 206.86 | 98.02 | 155.76 | 34.32 | 15.09 | −13.74 |
| AAPC(%) | 3.68* | 2.23* | 3.08* | 0.96* | 0.46* | −0.47* |
| 95%CI | 3.60, 3.77 | 2.21, 2.25 | 3.00, 3.16 | 0.94, 0.98 | 0.41, 0.52 | −0.50, −0.45 |
| Female | | | | | | |
| 1990 | 105.73 | 674.30 | 185.61 | 254.66 | 282.24 | 329.74 |
| 2021 | 348.60 | 1391.11 | 501.84 | 353.80 | 328.08 | 301.57 |
| Change (%) | 229.71 | 106.30 | 170.37 | 38.93 | 16.24 | −8.54 |
| AAPC(%) | 3.91* | 2.36* | 3.26* | 1.06* | 0.47* | −0.29* |
| 95%CI | 3.80, 4.01 | 2.33, 2.39 | 3.15, 3.37 | 1.05, 1.08 | 0.36, 0.57 | −0.31, −0.28 |

IHD, Ischemic heart disease; ASIR, Age-standardized incidence rate; AAPC, Average annual percentage of change; CI, confidence interval; * P<0.001.

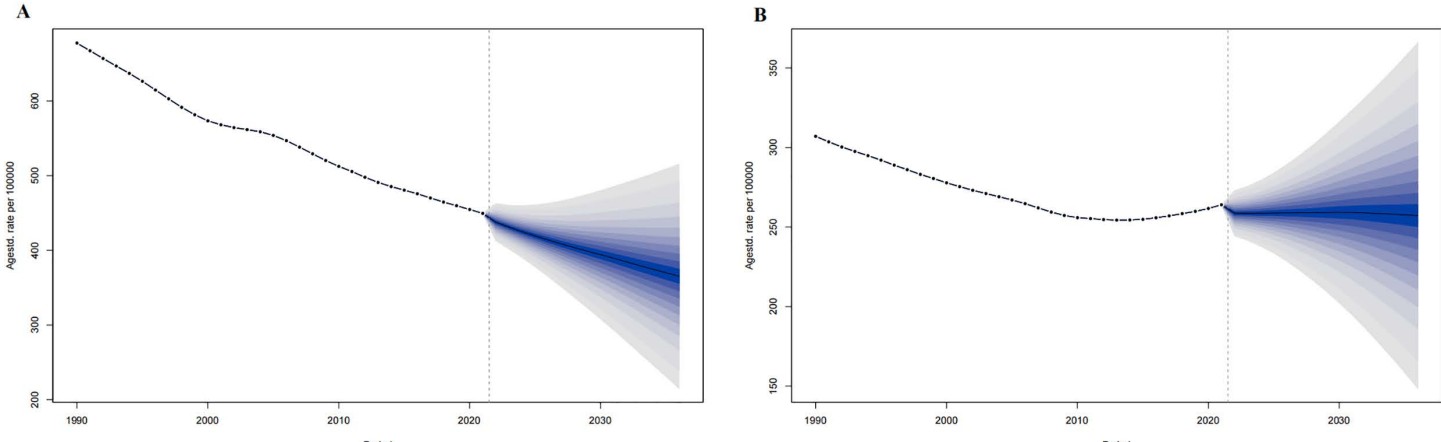

**Fig 1. The ASIR prediction of IHD in the world from 2022 to 2036.** ASIR prediction in Global male (A), Global female (B). The points prior to the dashed line (representing 2021) indicate the observed values, the solid line following represents the projected ASIR from 2022 to 2036. The blue shadow indicates 95% UI. ASIR, Age-standardized incidence rate. UI, the 95% uncertainty intervals.

**Table 2. Prevalence of IHD in China and global, 1990-2021.**

| Gender | Number of cases (10,000) | | prevalence rate (per 100,000) | | ASPR (per 100,000) | |
|---|---|---|---|---|---|---|
| | China | Global | China | Global | China | Global |
| Both | | | | | | |
| 1990 | 1950.55 | 11216.95 | 1657.98 | 2103.06 | 2526.44 | 2904.72 |
| 2021 | 6333.13 | 25427.63 | 4451.34 | 3222.21 | 3042.35 | 2946.38 |
| Change (%) | 224.68 | 126.69 | 168.48 | 53.22 | 20.42 | 1.43 |
| AAPC(%) | 3.85* | 2.67* | 3.23* | 1.38* | 0.60* | 0.03 |
| 95%CI | 3.83, 3.88 | 2.65, 2.69 | 3.20, 3.26 | 1.36, 1.40 | 0.58, 0.62 | 0.00, 0.06 |
| Male | | | | | | |
| 1990 | 1064.87 | 6498.05 | 1754.78 | 2419.46 | 2838.18 | 3688.47 |
| 2021 | 3357.19 | 14530.77 | 4610.87 | 3669.95 | 3379.15 | 3610.24 |
| Change (%) | 215.27 | 123.62 | 162.76 | 51.68 | 19.06 | −2.12 |
| AAPC(%) | 3.78* | 2.63* | 3.17* | 1.36* | 0.56* | −0.07* |
| 95%CI | 3.75, 3.80 | 2.62, 2.64 | 3.12, 3.22 | 1.35, 1.37 | 0.53, 0.59 | −0.07, −0.07 |
| Female | | | | | | |
| 1990 | 885.68 | 4718.90 | 1554.86 | 1782.14 | 2235.32 | 2250.59 |
| 2021 | 2975.94 | 10896.86 | 4284.14 | 2771.35 | 2724.16 | 2357.61 |
| Change (%) | 236.01 | 130.92 | 175.53 | 55.51 | 21.87 | 4.76 |
| AAPC(%) | 3.95* | 2.71* | 3.32* | 1.41* | 0.63* | 0.15* |
| 95%CI | 3.90, 4.00 | 2.68, 2.75 | 3.27, 3.37 | 1.37, 1.45 | 0.58, 0.67 | 0.10, 0.19 |

IHD, Ischemic heart disease; ASPR, Age-standardized prevalence rate; AAPC, Average annual percentage of change; CI, confidence interval; * $P < 0.001$..

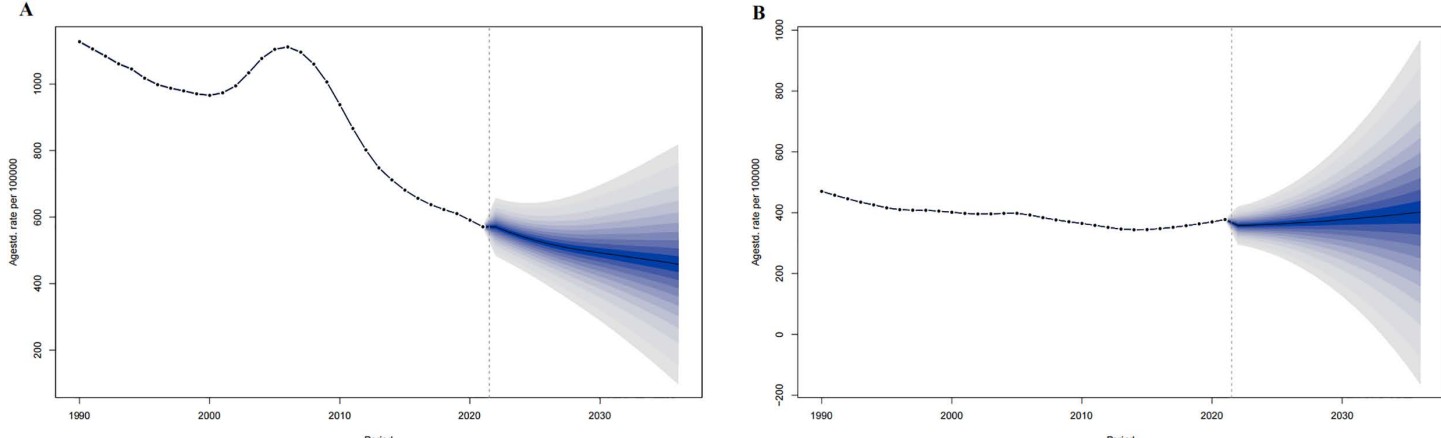

**Fig 2. The ASIR prediction of IHD in China from 2022 to 2036.** ASIR prediction in China male (A), China female (B). The points prior to the dashed line (representing 2021) indicate the observed values, the solid line following represents the projected ASiR from 2022 to 2036. The blue shadow indicates 95% UI. ASIR, Age-standardized incidence rate. UI, the 95% uncertainty intervals.

**Table 3. Mortality of IHD in China and the world, 1990-2021.**

| Gender | Number of deaths (10,000) | | deaths rate (per 100,000) | | ASMR (per 100,000) | |
|---|---|---|---|---|---|---|
| | China | Global | China | Global | China | Global |
| Both | | | | | | |
| 1990 | 54.78 | 536.71 | 46.57 | 100.63 | 94.14 | 158.90 |
| 2021 | 195.69 | 899.16 | 137.54 | 113.94 | 110.91 | 108.73 |
| Change (%) | 257.23 | 67.53 | 195.34 | 13.22 | 17.81 | −31.57 |
| AAPC(%) | 4.16* | 1.72* | 3.54* | 0.43* | 0.49* | −1.20* |
| 95%CI | 3.96, 4.37 | 1.54, 1.90 | 3.31, 3.78 | 0.28, 0.58 | 0.23, 0.75 | −1.37, −1.03 |
| Male | | | | | | |
| 1990 | 28.10 | 280.50 | 46.31 | 104.44 | 109.77 | 187.66 |
| 2021 | 108.82 | 500.27 | 149.46 | 126.35 | 148.40 | 136.84 |
| Change (%) | 287.26 | 78.35 | 222.74 | 20.98 | 35.19 | −27.08 |
| AAPC(%) | 4.46* | 1.91* | 3.84* | 0.65* | 0.95* | −1.02* |
| 95%CI | 4.24, 4.68 | 1.74, 2.07 | 3.62, 4.07 | 0.50, 0.80 | 0.60, 1.31 | −1.21, −0.83 |
| Female | | | | | | |
| 1990 | 26.68 | 256.22 | 46.85 | 96.76 | 84.41 | 134.50 |
| 2021 | 86.87 | 398.90 | 125.05 | 101.45 | 86.10 | 85.32 |
| Change (%) | 225.60 | 55.69 | 166.92 | 4.85 | 2.00 | −36.57 |
| AAPC(%) | 3.86* | 1.48* | 3.18* | 0.17* | 0.03 | −1.44* |
| 95%CI | 3.62, 4.09 | 1.32, 1.64 | 2.85, 3.51 | 0.02, 0.33 | −0.24, 0.29 | −1.60, −1.28 |

IHD, Ischemic heart disease; ASMR, Age-standardized mortality rate; AAPC, Average annual percentage of change; CI, confidence interval; * $P < 0.001$.

### 3.2. The burden of IHD in China in 1990–2021 and prediction in 2022–2036

**3.2.1. Incidence of IHD in China.** In China, new IHD cases rose from 2,301,600–7,304,600 between 1990 and 2021. The growth rate (217.37%) and AAPC (3.80%, 95% CI: 3.67%~3.92%, P<0.001) surpassed global figures. The incidence

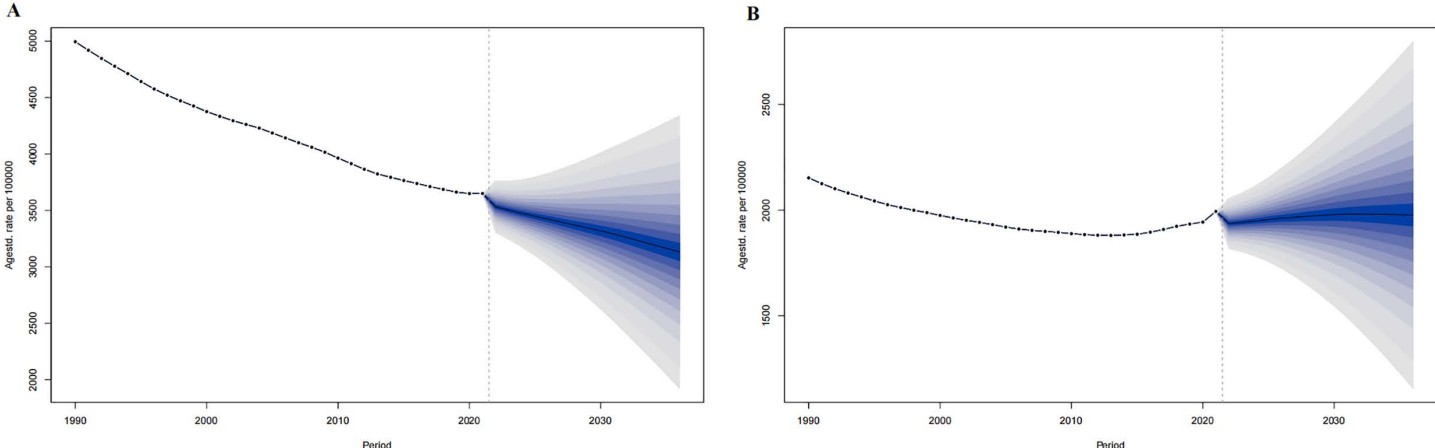

**Fig 3. The ASPR prediction of IHD in the world from 2022 to 2036.** ASPR prediction in Global male (A), Global female (B). The points prior to the dashed line (representing 2021) indicate the observed values, the solid line following represents the projected ASPR from 2022 to 2036. The blue shadow indicates 95% UI. ASPR, Age-standardized prevalence rate. UI, the 95% uncertainty intervals.

rate and ASIR rose from 195.64 per 100,000 and 315.31 per 100,000 in 1990 to 513.41 per 100,000 and 365.67 per 100,000 in 2021, respectively. Their growth rates (162.43% for incidence rate; 15.97% for ASIR) and AAPC (3.17%, 95% CI: 3.10%~3.23% for incidence rate; 0.49%, 95% CI: 0.43%–0.55% for ASIR) and (0.49%, 95% CI: 0.43%~0.55%) were notably higher than global figures, with P<0.001 for all aforementioned AAPC values. (Table 1)

For 2022–2036 in China, the ASIR in men is predicted to decrease from 177.17 per 100,000 in 2022 to 151.69 per 100,000 in 2036, and the ASIR in women will rise from 162.78 in 2022 per 100,000 to 177.24 per 100,000 in 2036. (Fig 4)

**3.2.2. Prevalence of IHD in China.** In China, the number of IHD cases rose from 19,505,500–63,331,300 between 1990 and 2021. The growth rate (224.68%) and AAPC (3.85%, 95% CI: 3.83%~3.88%, P<0.001) surpassed global figures. The prevalence rate and ASPR rose from 1657.98 per 100,000 and 2526.44 per 100,000 in 1990 to 4451.34

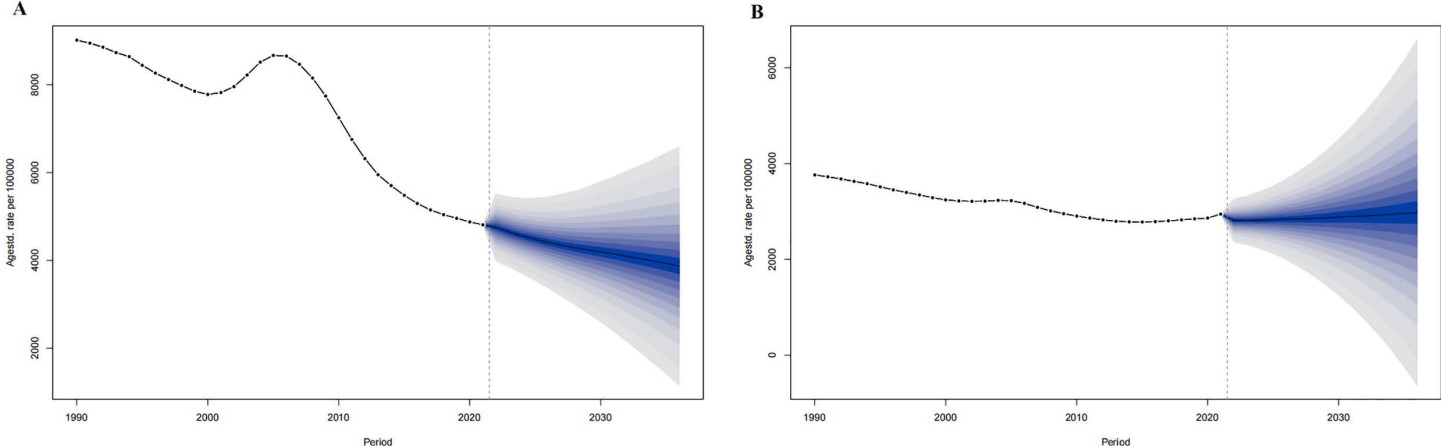

**Fig 4. The ASPR prediction of IHD in China from 2022 to 2036.** ASPR prediction in China male (A), China female (B). The points prior to the dashed line (representing 2021) indicate the observed values, the solid line following represents the projected ASPR from 2022 to 2036. The blue shadow indicates the 95% UI. ASPR, Age-standardized prevalence rate. UI, the 95% uncertainty intervals.

per 100,000 and 3042.35 per 100,000 in 2021, respectively. In 1990, China's prevalence rate and ASPR were lower than global figures; by 2021, both surpassed global figures. Their growth rates (168.48% for prevalence rate; 20.42% for ASPR) and AAPC (3.23%, 95% CI: 3.20%~3.26% for prevalence rate; 0.60%, 95% CI: 0.58%~0.62% for ASPR) were higher than global figures, with P<0.001 for all aforementioned AAPC values. (Table 2)

For 2022–2036 in China, the ASPR in men is predicted to decrease from 1525.51 per 100,000 in 2022 to 1436.03 per 100,000 in 2036, and the ASPR in women will rise from 1377.95 per 100,000 in2022 to 1641.64 per 100,000 in 2036. (Fig 5)

**3.2.3  Mortality of IHD in China.**  In China, the number of IHD deaths rose from 547,800–1956,900. The growth rate (257.23%) and AAPC (4.16%, 95% CI: 3.96%~4.37%, P<0.001) surpassed global figures. The mortality rate and ASMR rose from 46.57 per 100,000 and 94.14 per 100,000 in 1990 to 137.54 per 100,000 and 110.91 per 100,000 in 2021, respectively. In 1990, China's mortality rate and ASMR were lower than global figures; by 2021, both surpassed global figures. Their growth rates (195.34% for mortality rate; 17.81% for ASMR) and AAPC (3.54%, 95% CI: 3.31%~3.78% for mortality rate; 0.49%, 95% CI: 0.23%~0.75% for ASMR) were higher than global figures, with P<0.001 for all aforementioned AAPC values. (Table 3)

For 2022–2036 in the world, the ASMR in both men and women are predicted to decrease from 70.77 per 100,000 and 119.33 per 100,000 in 2022 to 56.43 per 100,000 and 65.01 per 100,000 in 2036, respectively. (Fig 6)

### 3.3.  Comparison of the burden of IHD among different gender and age groups in China and the world

**3.3.1.  Comparison of IHD incidence.**  In both China and the world, men showed higher new IHD cases, incidence rates, and ASIR compared to women, yet lower corresponding growth rates and AAPC compared to women. (Table 1, Fig 1). New IHD cases and incidence rates in China and the world rose with age. In both China and the world, new IHD cases in all age groups showed an upward trend from 1990 to 2021, with the 80＋age group having the most cases and the＜20 age group having the fewest until 2021. For incidence rate, all age groups showed an upward trend from 1990 to 2021 in China, with the 80＋age group having the highest and the＜20 age group having the lowest until 2021; Globally, all age groups showed a downward trend, with the 80＋age group having the highest and the＜20 age group having the lowest until 2021. (Fig 7)

**3.3.2.  Comparison of IHD prevalence.**  In both China and the world, men showed higher IHD cases, prevalence rates, and ASPR compared to women, yet lower corresponding growth rates and AAPC compared to women. (Table 2,

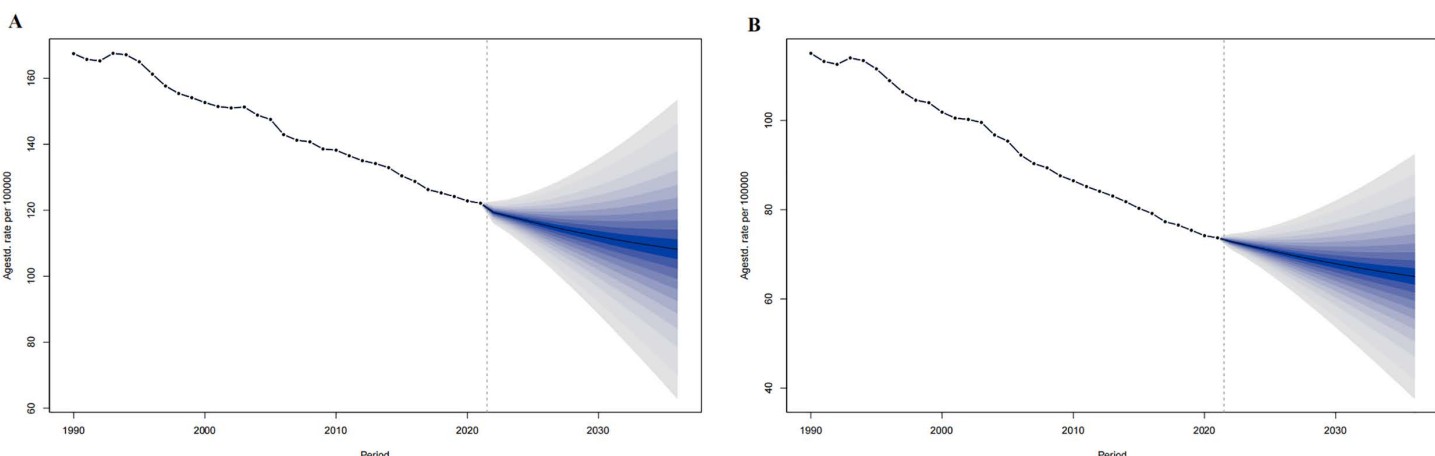

**Fig 5.  The ASMR prediction of IHD the world from 2022 to 2036.** ASMR prediction in Global male (A), Global female (B). The points prior to the dashed line (representing 2021) indicate the observed values, the solid line following represents the projected ASMR from 2022 to 2036. The blue shadow indicates the 95% UI. ASMR, Age-standardized prevalence rate. UI, the 95% uncertainty intervals.

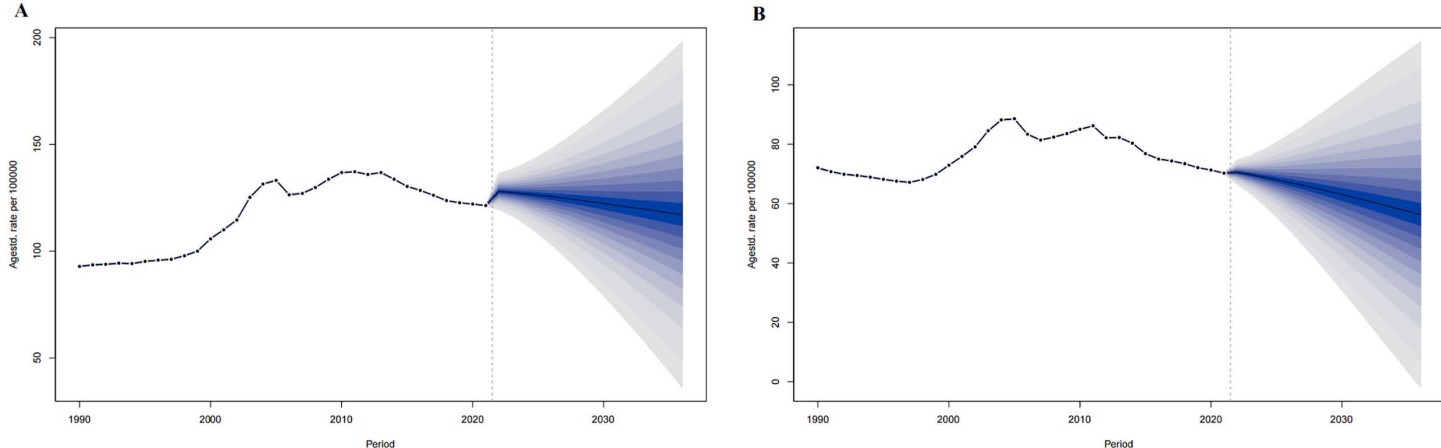

**Fig 6. The ASMR prediction of IHD in China from 2022 to 2036.** ASMR prediction in China male (A), China female (B). The points prior to the dashed line (representing 2021) indicate the observed values, the solid line following represents the projected ASMR from 2022 to 2036. The blue shadow indicates the 95% UI. ASMR, Age-standardized prevalence rate. UI, the 95% uncertainty intervals.

Fig 3). IHD cases and prevalence rates in China and the world rose with age. In both China and the world, IHD cases in all age groups showed an upward trend from 1990 to 2021, with the 80 + age group having the most cases and the < 20 age group having the fewest until 2021. For prevalence rate, all age groups showed an upward trend from 1990 to 2021 in China, with the 80 + age group having the highest and the < 20 age group having the lowest until 2021; Globally, ≥ 70 age groups showed a downward trend and ≤70 age groups showed an upward trend (except <20 age group), with the 80 + age group having the highest and the < 20 age group having the lowest until 2021. (Fig 8)

3.3.3. **Comparison of IHD mortality.** In both China and the world, excluding the Chinese mortality rate in 1990, men showed higher deaths, mortality rates, and ASMR of IHD than women. Corresponding growth rates and AAPC, except for global ASMR, were higher in men than in women. (Table 3, Fig 5). IHD deaths and mortality rates in China and the world rose with age. In both China and the world, IHD deaths in all age groups showed an upward trend from 1990 to 2021, with the 80 + age group having the most cases and the < 20 age group having the fewest until 2021. For mortality rate, five age groups showed an upward trend in China, excluding the three groups in the 54–70 age range, with the 80 + age groups having the highest and the < 20 age group having the lowest until 2021; Globally, all age groups showed a downward trend, with the 80 + age group having the highest and the < 20 age group having the lowest until 2021. (Fig 9)

## 4. Discussion

Based on GBD 2021, we systematically compared the epidemiological characteristics of IHD in China and the world from 1990 to 2021, and predicted the trend by sex in the next 15 years. In 1990, China's IHD burden was lower than global figures, but its growth rate was faster, by 2021, except for the ASIR, all other values in China were higher than global figures. According to the prediction results, we found that after 2021, the burden of IHD in China will be higher than that in global. In addition, we found that there were significant age differences and gender differences in the burden of IHD in China. High incidence, prevalence, and mortality show an upward trend with increasing age and it was predicted that the ASMR for IHD will decline in China and the world from 2022 to 2036, and for ASIR and ASPR, men's will decline while women's will increase.

IHD has already become the second leading cause of death in China, bringing huge burden to social and economic development [13]. This study found that China's IHD burden increased year by year since 1990, and exceeded global figures in 2021. Common pathogenic factors of IHD include hypertension, hypercholesterolemia, diabetes, etc [14]. In

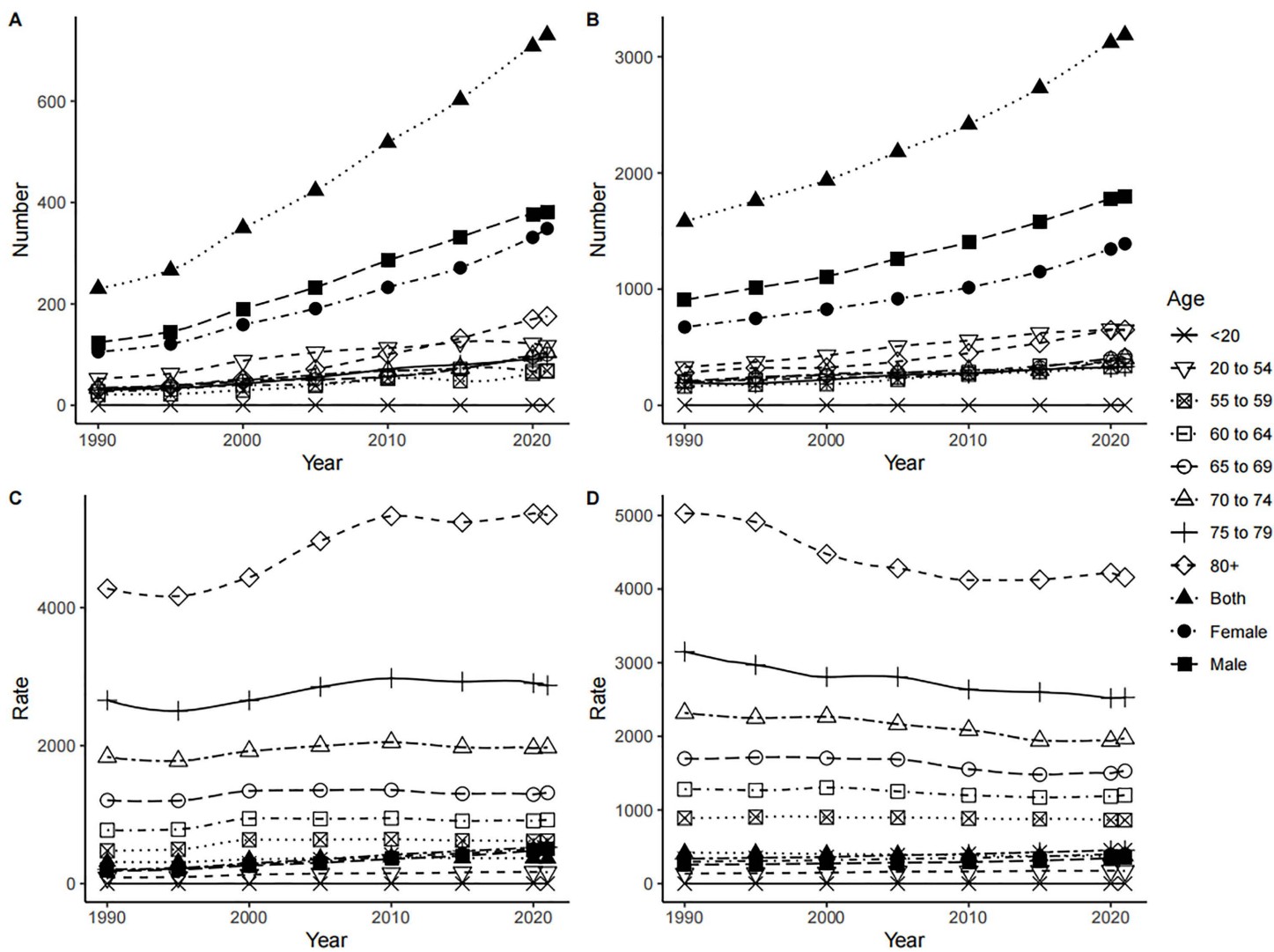

**Fig 7. Incidence of IHD in China and the world by age and sex from 1990 to 2021.** New IHD cases in China (A) and Global (B), incidence rate in China (C) and Global (D).

addition, some studies have shown that there is a positive correlation between meat intake and the incidence of IHD [15,16]. Some studies have shown that, with the development of the economy, the Chinese diet has become dominated by high fat and high calorie intake [17]. Meanwhile, data from the World Health Organization indicates that the risk of developing IHD in people with normal blood pressure and cholesterol levels is at least 16% lower than those with hypertension and hypercholesterolemia [18]. Therefore, it is necessary to advocate a light diet. The higher mortality and burden of IHD in China may be related to late diagnosis due to insufficient public awareness of the disease, unequal distribution of healthcare resources, and lack of systematic screening. In addition, China's medical resources were more concentrated in large cities and developed coastal areas. This may make it difficult for residents in some areas to access timely and effective IHD screening and treatment.

This study found that there were gender differences in the burden of IHD, which is consistent with previous research results. The incidence, prevalence and mortality of IHD in Chinese men were higher than those in women.

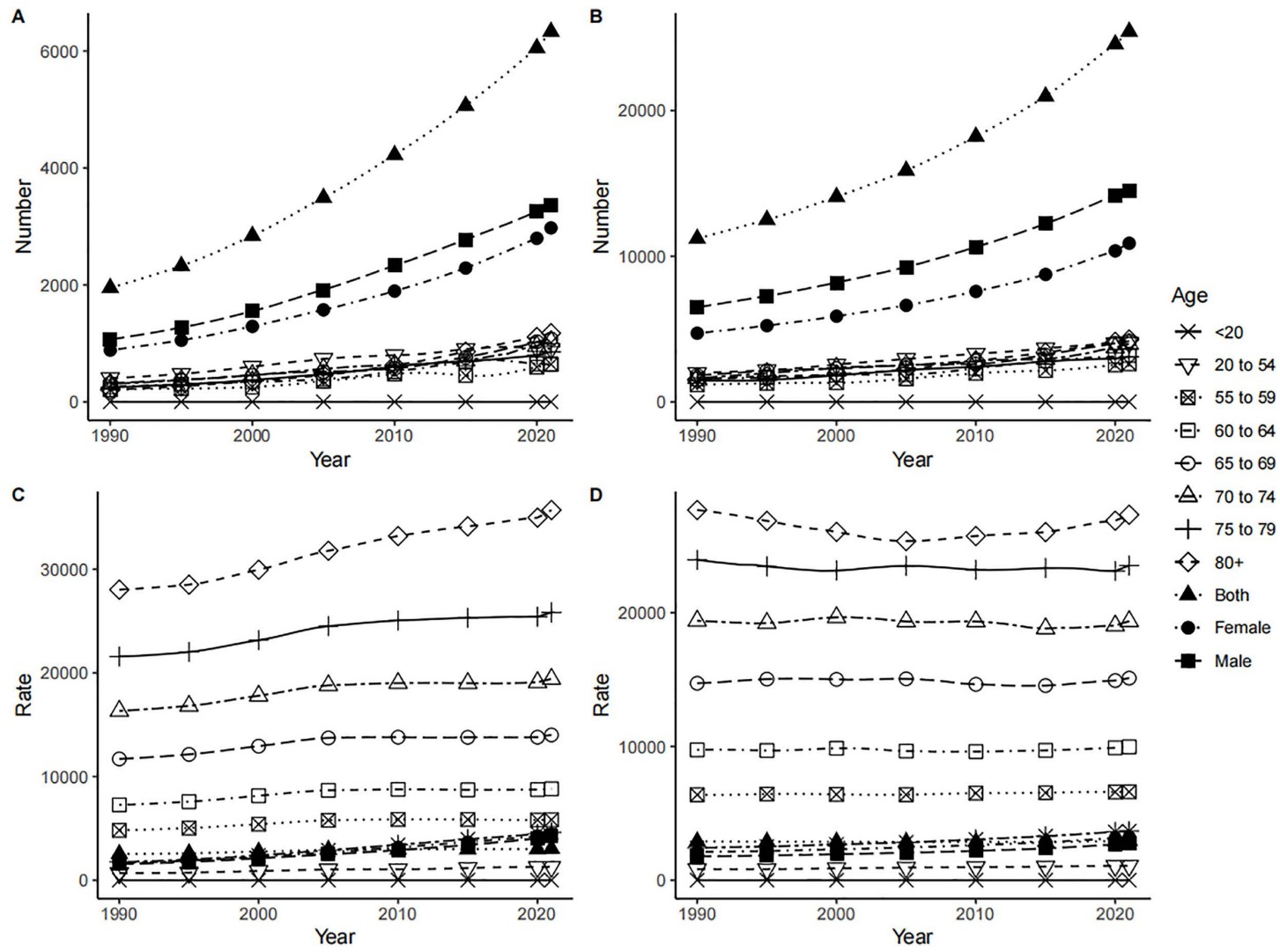

**Fig 8. Prevalence of IHD in China and the world by age and sex from 1990 to 2021.** IHD cases in China (A) and Global (B), prevalence rate in China (C) and Global (D).

Several studies also showed that male residents were more vulnerable to IHD than female residents [19,20]. A high proportion of men have habits such as smoking, excessive drinking [14], and high-fat diets, which can increase their risk of developing disease like hypertension and hypercholesterolemia. Additionally, women have multiple protective factors for lower IHD burden. Sex hormones, particularly the synergistic action of estrogen and progesterone in premenopausal women, help regulate lipids, maintain vascular elasticity, and slow atherosclerosis [21]. They have more subcutaneous fat, unlike men's predominant visceral fat that releases harmful factors. Women also pay greater attention to health management, such as regular check-ups. These, combined with hormonal reduction of thrombosis risk, may lower their IHD burden, especially pre-menopause. Therefore, men should be guided to abandon unhealthy lifestyles, including behaviors such as smoking, poor dietary choices, and neglect of regular health check-ups. Their awareness of prevention strategies and self-care methods for cardiovascular diseases (such as IHD) should be enhanced.

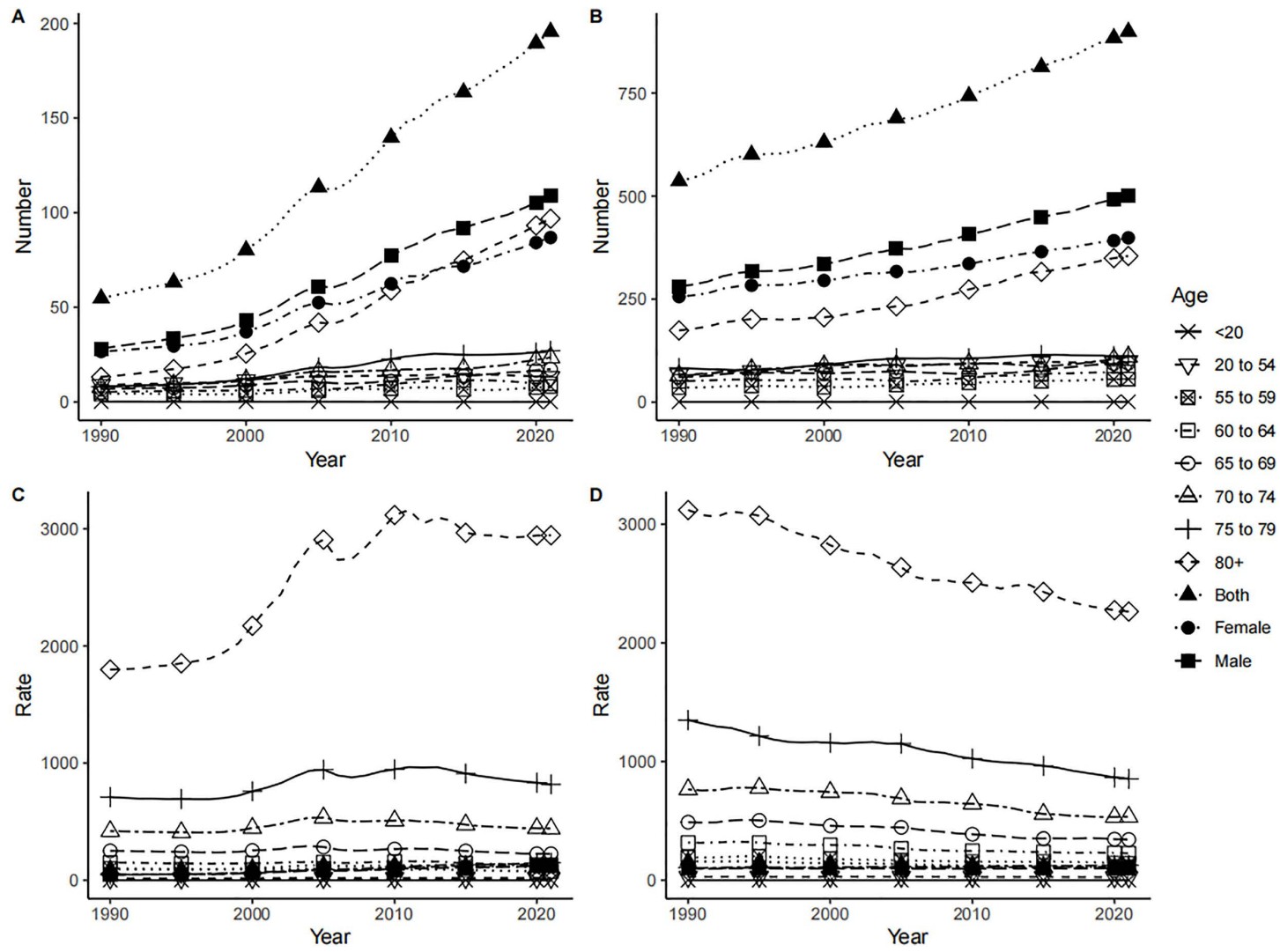

**Fig 9. Mortality of IHD in China and the world by age and sex from 1990 to 2021.** IHD deaths in China (A) and Global (B), mortality rate in China (C) and Global (D).

However, we found that the AAPC of IHD in China was lower in men than in women. This may because women are more susceptible to IHD under the influence of certain risk factors. For example, the risk of developing the disease in women with diabetes is 5–7 times higher than that in healthy women [14,22], and excessive physical activity also makes women more likely to suffer from IHD [23]. In addition, studies have shown that a series of female-specific risk factors such as menopause, gestational diabetes mellitus, and the use of oral contraceptives can also increase women's risk of developing cardiovascular diseases [24,25]. Enhancing attention to cardiovascular diseases in women and implementing gender-specific stratified management of cardiovascular risk factors, which is crucial for accurately reducing the burden of cardiovascular diseases in the female population..

This study also found that there were age differences in the burden of IHD, especially in China. From 1990 to 2021, although the incidence rate, prevalence rate and mortality rate of ≥70 age groups worldwide have been decreasing year by year, and the incidence rate and prevalence rate of ≤70 age groups have shown and overall upward trend, the

incidence, prevalence and deaths of IHD in China and global increased with age [16]. Therefore, despite the improvement of medical care level can prolong the survival period of patients with heart disease, with the growth of population and the aggravation of population aging, the absolute number of deaths from IHD continues to increase. This study showed that from 1990 to 2021, the incidence rate, prevalence rate, and mortality rate of IHD of ≥70 age groups in China all showed an upward trend, and their IHD burden increased significantly. As we age, coronary arteries and microvessels become less functional and resilient and may be altered by myocardial perfusion injury [26]. Therefore, in the face of the increasingly serious aging level and severe medical society and socio-economic challenges, it is urgent to achieve a healthy aging society.

Meanwhile, In 2021, the incidence and prevalence of IHD among Chinese individuals aged 20–54 were higher than those recorded in 1990, which was similar to previous study. It found that in recent years, young and middle-aged people with IHD have factors such as overweight, smoking, abnormal lipid metabolism, high blood uric acid [20]. As a result, the incidence, prevalence and mortality of IHD in China are on the rise in young people [19,27]. The study suggest that primary healthcare institutions should focus on strenthening health education activities for young and middle-aged people, improving their awareness of tertiary prevention knowledge regarding IHD, and reducing the harm caused by the disease [28].

Based on the projected dara for IHD in China and the world over the next 15 years, from 2022 to 2036, the ASMR will continue to decline, and for ASIR and ASPR, men's will decline while women's will increase. With the improvement of living standard and education level, it may be related to that Chinese people were paying more attention to their health and increased their intake of vegetables and fiber [1]. Nevertheless, in the next few years, with China's aging population, accelerated pace of life, and changes in diet, the number of cases and deaths of IHD will continue to increase, and the disease burden will be significantly higher than global figures. Therefore, China should pay more attention to IHD, strengthen the primary and secondary prevention of IHD, promote the popularization of IHD science effectively, raise residents' awareness of IHD, and strengthen the level of diagnosis and treatment of IHD to reduce the burden.

There were some limitations in our study. The analysis of the disease burden of IHD in this study is currently limited to the national level.It has not yet extended to the calculation of the disease burden in populations at local levels such as provinces, cities and countries, nor has it included a comparative exploration between urban and rural areas.This suggest that it is necessary to further deepen research on the disease burden at the local level in China in the future. In addition,the initial IHD data are also affected by factors such as the quantity, quality and correcting methods of the data used for modeling.To ensure the prediction results are as precise as possible, this study only projected changes over the next 15 years.

## 5. Conclusion

Based on GBD 2021 data, the burden of IHD was analyzed and predicted, which not only provided a reference for public policymakers to formulate IHD prevention but also provided warnings for patients to increase health education. Because of aging population and shortage of medical resources, regular physical examination and high-risk factor intervention for key groups should be strengthened. Men should pay more attention to their mental health, avoid excessive smoking, drink less, and reduce high-fat diet; women should try to avoid excessive physical labor and pay attention to pregnancy diet. In addition, due to the increasing incidence and prevalence of IHD among young and middle-aged people, health education for these people will help improve patients' compliance with treatment, such as informing them of reasonable diet, exercise and weight loss, limiting smoking and drinking, etc., to reduce the burden of IHD in China and the world.

## Supporting information

**S1 Dataset. Dataset for Table1–3.**
(CSV)

**S2 Dataset. Dataset for Figures1–4.**
(CSV)

**S3 Dataset. Dataset for Figures5–6.**
(CSV)

**S4 Dataset. Dataset for Figures7.**
(CSV)

**S5 Dataset. Dataset for Figures8.**
(CSV)

**S6 Dataset. Datasets for Figures9.**
(CSV)

## Author contributions

**Conceptualization:** Jiale Wu, Yizhong Yan.

**Data curation:** Jiale Wu, Jinchan Zhai, Xingyun Zhu, Quan Hao, Jiaqi Yang, Yunhua Hu.

**Funding acquisition:** Yizhong Yan.

**Methodology:** Jinchan Zhai.

**Project administration:** Qiang Niu.

**Resources:** Xiaoju Li, Qiang Niu.

**Supervision:** Xiaoju Li, Yunhua Hu, Yizhong Yan.

**Validation:** Xingyun Zhu, Quan Hao, Jiaqi Yang, Xiaoju Li, Qiang Niu.

**Visualization:** Jiale Wu, Xingyun Zhu, Jiaqi Yang.

**Writing – original draft:** Jiale Wu, Jinchan Zhai, Xingyun Zhu.

**Writing – review & editing:** Jiale Wu, Jinchan Zhai, Quan Hao, Yunhua Hu, Yizhong Yan.

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
