## [Decision Letter · Decision Letter 0]

2 Oct 2025

Dear Dr. Yan,

We look forward to receiving your revised manuscript.

Kind regards,

Redoy Ranjan, MBBS, MRCSEd, Ch.M., MS (CV&TS), FACS

Academic Editor

PLOS ONE

Journal Requirements:

“The Shihezi University High-level Talents Program (RCZK2021B28). The Science and Technology Planning Project of Xinjiang Production and Construction Corps (2023AB049). The Shihezi University self-funded project (ZZZC202125). Tianshan Young Talent Scientific and Technological Innovation Team: Innovative Team for Research on Prevention and Treatment of High-incidence Diseases in Central Asia (2023TSYCTD0020).”

4. We note that your Data Availability Statement is currently as follows: “All relevant data are within the manuscript and its Supporting Information files.”

5. Please note that funding information should not appear in any section or other areas of your manuscript. We will only publish funding information present in the Funding Statement section of the online submission form. Please remove any funding-related text from the manuscript.

Reviewer's Responses to Questions

**Comments to the Author**

1. Is the manuscript technically sound, and do the data support the conclusions?

Reviewer #1: Yes

Reviewer #2: Yes

Reviewer #3: Yes

2. Has the statistical analysis been performed appropriately and rigorously?

Reviewer #1: Yes

Reviewer #2: Yes

Reviewer #3: Yes

3. Have the authors made all data underlying the findings in their manuscript fully available?

Reviewer #1: Yes

Reviewer #2: Yes

Reviewer #3: Yes

4. Is the manuscript presented in an intelligible fashion and written in standard English?

Reviewer #1: Yes

Reviewer #2: Yes

Reviewer #3: Yes

Reviewer #1: Quite interesting data regarding prevalence and incidence of ischaemic heart disease in a certain population compared to the rest of the world highlighting the complicated nature of this disease and the role of lifestyle and genetics in the incidence and management of ischaemic heart disease. Moreover, the remaining high incidence in the vast Chinese population highlights the need for more measures to be taken.

Reviewer #2: The article is very detailed and well-written. A study utilized Global Burden of Disease (GBD) 2021 data to compare the burden of ischemic heart disease (IHD) in China and the world from 1990 to 2021, with projections extending to 2036. Join point regression was employed to estimate annual changes in burden (Annual Percent Change, AAPC), and Bayesian age–period–cohort (BAPC) modeling was used for forecasts; results indicate that, from 1990 to 2021, the IHD burden in China increased and progressed more rapidly than the global trend. There are a few comments that could potentially improve the manuscript.

1-In the data source section, although the GBD 2021 source is cited, a brief explanation of sample size and data collection methods can aid in assessing generalizability.

2-All articles submitted to this journal require a code of ethics. For more information, visit

https://journals.plos.org/plosone/s/submission-guidelines#loc-human-subjects-research

3-In the data analysis section, a more precise description of how the data from 1990–2021 are synthesized using the Joinpoint and Bayesian age–period–cohort (BAPC) models, together with the key parameters (e.g., the initial joinpoint, the number of joinpoints), would be informative.

4-Some tables (e.g., Table 3) should clearly state in the text when and where statistical differences are significant (significance symbols should appear with all relevant values). It is advisable that the manuscript be written so that the main points are still conveyed through the text if the table is not visible. In addition, the confidence intervals (CIs) are not displayed correctly in some sections; this requires more careful editing, e.g., (3.67, 3.92).

5-The discussion suggests that it be strengthened with additional evidence and appropriate references to support the interpretations and claims. It also points to more practical policies such as targeted screening programs, lifestyle interventions, and gender-specific recommendations in line with the findings.

Reviewer #3: One misspelling in a section title is noted in the attached review. The text in the article is occasionally slightly informal and narrative, but still well constructed. This is very much like other articles now commonly being produced by researchers, due to changes in research resources and support. Reviews of our new databases are essential to the field and should not at all deemed too repetitive by publishers, even though they can be. They are essential to the changes now happening in AI, data analysis and management services in health care. I just reviewed one other item of similar methods, and purpose in this industry.

**Do you want your identity to be public for this peer review?** For information about this choice, including consent withdrawal, please see our Privacy Policy

Reviewer #1: **Yes: ** Afendoulis Dimitrios

Reviewer #2: No

Reviewer #3: **Yes: ** Brian L Altonen

---

## [Author Response · Author response to Decision Letter 1]

18 Oct 2025

Response to Reviewers

We would like to express our sincere gratitude to all the reviewers for their valuable comments and constructive suggestions, which have been of great help in improving our manuscript. We have carefully addressed each comment, and the detailed responses are as follows:

Response to Reviewer #1

Thank you for your positive comment on our study regarding the prevalence and incidence of ischemic heart disease (IHD) in specific populations. We are pleased that our research has highlighted the complexity of this disease and the roles of lifestyle and genetics, as well as the high incidence in China's large population, which indeed underscores the need for more intervention measures. We will continue to pay attention to this issue and explore more effective strategies in future research.

Response to Reviewer #2

We are grateful for your comprehensive and insightful comments, as well as your positive evaluation of the manuscript's detail and writing. We have carefully revised the manuscript according to your suggestions:

1.Regarding the data source section: We have added a brief description of the sample size and data collection methods of GBD 2021 in the revised manuscript. Specifically, GBD 2021 provided epidemiological data for 204 countries or regions, 371 diseases, and 88 risk factors. This addition is intended to help assess the generalizability of the study.

2.Regarding ethical guidelines: The data for this study were sourced from a publicly available database, requiring no ethical approval or informed consent. No ethical approval and informed consent were required because of the public availability of GBD and no identifiable information was included in the analyses.

3.Regarding the data analysis section: We have more accurately described how the joinpoint and Bayesian age-period-cohort (BAPC) models were used to synthesize data from 1990 to 2021, and have also added explanations of key parameters such as the initial joinpoint and the number of joinpoints to provide more detailed information.

4.Regarding tables and confidence intervals: We have checked all tables (including Table 3) and explicitly indicated statistical significance in the text, with all relevant values marked with significance symbols. We have also ensured that confidence intervals (e.g., (3.67, 3.92)) are correctly displayed throughout the manuscript through careful editing. Additionally, we have revised the manuscript to ensure that the key points can still be conveyed through the text even if the tables are not visible.

5.Regarding the discussion section: We have strengthened the discussion by adding additional evidence and appropriate references to support our interpretations and claims.

Response to Reviewer #3

Thank you for pointing out the spelling error in the chapter title, which we have corrected in the revised manuscript. We also appreciate your comment on the writing style; we have reviewed the text to make it more formal while maintaining its structural clarity. We agree with your view that the review of new databases is crucial for the field, especially in the context of changes in artificial intelligence, data analysis, and management services in healthcare. We will continue to contribute to the advancement of research in this area.

Once again, we would like to thank all the reviewers for their time and efforts in reviewing our manuscript. We hope that the revised version meets your expectations.

---

## [Decision Letter · Decision Letter 1]

4 Nov 2025

Epidemiological characteristics of ischemic heart disease: a comparative study between China and the world from 1990 to 2021 and prediction to 2036

PONE-D-25-41345R1

Dear Dr. Yan,

We’re pleased to inform you that your manuscript has been judged scientifically suitable for publication and will be formally accepted for publication once it meets all outstanding technical requirements.

Kind regards,

Redoy Ranjan, MBBS, MRCSEd, Ch.M., MS (CV&TS), FACS

Academic Editor

PLOS ONE

Additional Editor Comments (optional):

Reviewers' comments:

Reviewer's Responses to Questions

**Comments to the Author**

Reviewer #1: All comments have been addressed

Reviewer #2: All comments have been addressed

Reviewer #3: All comments have been addressed

2. Is the manuscript technically sound, and do the data support the conclusions?

Reviewer #1: Yes

Reviewer #2: Yes

Reviewer #3: Yes

3. Has the statistical analysis been performed appropriately and rigorously?

Reviewer #1: Yes

Reviewer #2: Yes

Reviewer #3: Yes

4. Have the authors made all data underlying the findings in their manuscript fully available?

Reviewer #1: Yes

Reviewer #2: Yes

Reviewer #3: Yes

5. Is the manuscript presented in an intelligible fashion and written in standard English?

Reviewer #1: Yes

Reviewer #2: Yes

Reviewer #3: Yes

Reviewer #1: I have carefully reviewed the revised edition of the manuscript. Given the changes made incorporating the reviwers recommendations i suggest upon publication of the manuscript.

Reviewer #2: (No Response)

Reviewer #3: A comprehensive summary. In terms of details and approaches taken, very useful for developing the same sorts of research across different systems. In general I accepted this in its initial submission form, although there were occasional, minor improvements (punctuation and misspelling) that could be made. The most important value of this report/article, is that it provides approaches that are similar across the field, for other articles I reviewed, which should make this article very applicable to my new residency hire serving "review of stats" program.; The increasingly popular population level meta-analysis reviews, is lacking in certain foreign (non-USA, non-Western European) countries studies. Comparing results from different countries or nations' studies is most important in our current health care systems. Searching for places/settings where significant differences do not exist between two very different health care settings, can have important clinical implications for the researchers of those projects. Such a value is very underrepresented and/or not covered effectively in certain kinds of research.

**Do you want your identity to be public for this peer review?** For information about this choice, including consent withdrawal, please see our Privacy Policy

Reviewer #1: **Yes: ** Afendoulis Dimitrios

Reviewer #2: No

Reviewer #3: **Yes: ** Brian L Altonen

---

## [Editor Report · Acceptance letter]

PONE-D-25-41345R1

PLOS ONE

Dear Dr. Yan,

I'm pleased to inform you that your manuscript has been deemed suitable for publication in PLOS ONE. Congratulations! Your manuscript is now being handed over to our production team.

Kind regards,

on behalf of

Dr. Redoy Ranjan

Academic Editor

PLOS ONE